# Challenges in the Use of AI-Driven Non-Destructive Spectroscopic Tools for Rapid Food Analysis

**DOI:** 10.3390/foods13060846

**Published:** 2024-03-10

**Authors:** Wenyang Jia, Konstantia Georgouli, Jesus Martinez-Del Rincon, Anastasios Koidis

**Affiliations:** 1Institute for Global Food Security, School of Biological Sciences, Queen’s University Belfast, Belfast BT9 5DL, UK; wjia02@qub.ac.uk (W.J.); kgeorgouli01@qub.ac.uk (K.G.); 2Lawrence Livermore National Laboratory, Livermore, CA 94550, USA; 3Institute of Electronics, Communications and Information Technology, Queen’s University Belfast, Belfast BT3 9DT, UK; j.martinez-del-rincon@qub.ac.uk

**Keywords:** chemometrics, spectroscopy, challenges, research, validation, methodology, modelling, guidelines

## Abstract

Routine, remote, and process analysis for foodstuffs is gaining attention and can provide more confidence for the food supply chain. A new generation of rapid methods is emerging both in the literature and in industry based on spectroscopy coupled with AI-driven modelling methods. Current published studies using these advanced methods are plagued by weaknesses, including sample size, abuse of advanced modelling techniques, and the process of validation for both the acquisition method and modelling. This paper aims to give a comprehensive overview of the analytical challenges faced in research and industrial settings where screening analysis is performed while providing practical solutions in the form of guidelines for a range of scenarios. After extended literature analysis, we conclude that there is no easy way to enhance the accuracy of the methods by using state-of-the-art modelling methods and the key remains that capturing good quality raw data from authentic samples in sufficient volume is very important along with robust validation. A comprehensive methodology involving suitable analytical techniques and interpretive modelling methods needs to be considered under a tailored experimental design whenever conducting rapid food analysis.

## 1. Introduction

Official methods adopted by regulatory authorities to protect a food product, define its authenticity, and tackle food adulteration are often chemical, ‘targeted’ methods that are designed to identify and quantify specific known compounds or markers (i.e., targets) as major or minor ingredients within the food. One of the targeted methods is chromatography, which produces specific signal peaks after sample extraction following in-column separation. Similar output is produced using direct mass spectrometry analysis, depending on the complexity of the sample and the analytical needs [1]. These peaks are then translated to specific analytes and quantified. The concern with these methods is that they are generally slow, non-environmentally friendly and require substantial initial capital investment and experienced staff to perform them. A new generation of rapid methods is emerging based on vibrational spectroscopy (mid-infrared, near-infrared, and Raman spectroscopy) and other sensors such as nuclear magnetic resonance spectroscopy and vision technology, fuelled by the rise in AI-based analytics. For instance, the coupling of multispectral/hyperspectral imaging and e-nose can rapidly produce a highly complex signal based on the properties and molecular structure of the foodstuffs that are analysed [2]. In contrast with targeted methods, in “untargeted” analysis, the outcome of the determination is not derived from the precise isolation and quantification of specific, known markers but from the chemical or molecular fingerprint of analytes extracted by the raw total spectrum, which is theoretically the sum of the signal from the sample’s constituents [3]. A common characteristic of all untargeted methods is that they provide a large number of signal peaks (in this case, absorbance or reflectance) coded as chemical information, presented in the form of numerous variables (wavelength or wavenumbers). The chemical information in the spectral data is often included within a lot of instrumental and environmental ‘noise’ [4]. The extraction and interpretation of this useful information are usually difficult and laborious. To process these multivariate data, a combination of mathematical and statistical techniques is used, commonly known as ‘chemometrics’. Nowadays, chemometrics, also referred to as “machine learning”, methods are used for developing prediction/decision models based on the knowledge acquired from calibration data using the principles of pattern recognition theory [5,6]. Recently, machine learning (ML) has been associated with the generic term ‘Artificial Intelligence’ (AI), although this is inaccurate as AI is a much broader term [7]. Thanks to these methods, information in multidimensional data can be automatically processed, elucidated, interpreted, and discriminated, resulting in advanced models that can authenticate, detect adulteration, and determine intrinsic quality parameters in foodstuffs [8]. Although not designed to completely replace the official regulatory ‘targeted’ methods, untargeted methods still have their place in the routine analysis and screening of foodstuffs. Especially after the pandemic, routine, remote, and process analysis for foodstuffs has been gaining attention and can provide more confidence for all stakeholders involved [9]. 

Currently, research on chemometrics and its application in the research field of food authenticity is thriving. The growth in the number of papers in the area is 700% in less than 20 years (Figure 1).

This figure depicts the rise in the number of research papers per year containing the keywords “chemometrics” and a combination of “chemometrics” and “Food”. The data show a significant increase over the years, illustrating the growing interest and research activity in this field. As the term ‘chemometrics’ is now replaced by ‘ML’ or ‘AI’, the actual studies related to the field of rapid analysis using the techniques described from 2022 and onwards could be even higher than demonstrated here (Figure 1). 

Nevertheless, new chemometrics techniques are being developed, and existing techniques that are sufficiently generic and based on a solid mathematical background are being improved to make them applicable to a number of problems. In the literature, chemometrics has been used as an essential tool for non-destructive tools to assess various scenarios such as beer classification, rice discrimination, and sensory evaluation [10]. In addition, the use of chemometrics has enabled the unified analysis of data derived from more than one analytical technique in a new field of “data fusion” and, consequently, more information about the analytes, although improved accuracy is not always guaranteed [11].

Current chemometric implementations in the literature are plagued by some common problems. One of them is the number of unique samples that the research studies are based on for developing or validating chemometric models, which evidently appears rather small. This constrains the use of more advanced AI techniques such as deep neural networks and transformers, which are state-of-the-art in machine learning but require hundreds if not thousands of samples to be effectively trained [12]. Another issue is the variable selection techniques that are based on visual inspection of the data and not algorithmic or knowledge-driven methodologies. In limited capacity in some research publications, advanced chemometrics methods are being applied in excess in an effort to add depth to the existing results and discussion. However, there are cases where they arguably do not add to the narrative; they end up vaguely supporting the original arguments made by the rest of the techniques and usually do not serve the objective of the study in which they appear.

Based on the above, this review paper aims to critically evaluate the methodology used by the authors of chemometric-based studies, highlight the analytical methods and current challenges, and identify some of the major reasons for inconsistencies in the practices used. Finally, this review will also provide practical suggestions for academic and industrial research to improve the technical approach taken in future scientific studies in order to provide innovative solutions, tackle overrated activities, and ensure food authenticity.

## 2. Challenges and Sources of Error in Current Research Studies

The research challenges in food analysis with regard to instrumental analytical techniques, experimental design, the application of ML/AI/chemometrics, and model validation, as well as analysts’ perspectives, are critically discussed below.

### 2.1. Analytical Methods

Among the various analytical techniques developed, molecular (including genomics and proteomics) and chromatographic methodologies have been used mainly for food authentication studies over the last few years [13]. As implied above, these targeted methods, such as high-performance liquid chromatography (HPLC), offer accurate, precise, and reliable results. However, despite their benefits, these methods are typically time-consuming and destructive and require expensive equipment and highly skilled personnel, which limits their application to the real-time screening of processes required by industry, where timely decisions are an essential requirement [14]. This general trend toward the use of molecular and chromatographic techniques poses fundamental questions regarding their adoption in an industrial setting. Specifically, the heterogeneity and physical nature of food samples usually require sample preparation, which is time-consuming, laborious, and destructive for classical targeted analytical techniques [15]. Besides sample preparation, the sample size used is small (<1 g) and therefore unlikely to be representative of the high-throughput production in industry, adding another angle that is important: the frequency of sampling, which by design is low when using targeted analytical methods. Most notably, these methods may allow other types of adulterants to remain undetected due to the targeted analysis of specific compounds or analytes of interest [16]. 

On the contrary, spectroscopic techniques such as vibrational spectroscopy enable rapid, easy, non-destructive, high-throughput, and relatively low-cost screening techniques with minimal sample preparation, with them being capable of online food analysis [17]. Here, the spectral information extracted is related to the molecular structure and, specifically, the vibrational behaviour of molecular bonds within a sample after the interaction of the sample with light [18]. Moreover, vibrational spectroscopy is characterised as a green analytical tool since it is an environmentally friendly option, minimising sample pretreatment with hazardous reagents and solvents [19]. Other techniques providing multivariate data such as hyper- and multispectral imaging have been developed as well to ensure food authenticity. As an advanced spectroscopy technique, hyperspectral imaging captures the spectrum of every pixel from the foodstuffs of interest, with it being developed regarding the authenticity of different types of foodstuffs, such as flour, meat, fruits, and oil. 

Although existing spectroscopic techniques coupled with chemometrics are capable of meeting food screening demands, a set of specific and challenging tasks should be overcome for their inclusion in real-world industrial applications and official methods (Figure 2). Therefore, research is needed to develop both reliable, state-of-the-art spectroscopic sensors and robust ML/AI chemometric models, as discussed in Section 2.3.

### 2.2. Experimental Design

The first consideration when designing a non-destructive spectroscopic analysis is the sampling plan, which can include certified or authentic food samples. Nowadays, most experimental studies that deal with food authentication are conducted to verify the labelling information about a specific food property (e.g., a certain geographic origin, variety, or composition) and to ensure its purity and quality. As a result, the absence of libraries of authentic samples (unlike the pharma industry where agencies such as Pharmacopeia exist) leads the literature authors to purchase food samples from unreliable sources, such as the retail market, for calibrating and validating their methods [20]. This compromises the integrity of the results since the authenticity of retail samples is not confirmed [21]. Additionally, the experimental calibration data used in food studies mainly focus on a specific source of variation for the development of classification models [22]. Furthermore, only a few studies capture some of the variability in the original samples introduced by different influential natural factors (e.g., climate, temperature, and geographical location) and other factors such as processing and, in most cases, storage conditions. The building of comprehensive databases of certified reference food materials for authenticity will allow for accurate and reliable food studies, which constitutes not only an important first step but also a considerable challenge [23].

To extend the use of non-destructive spectroscopic methods, the variability in classification models, from the acquisition of data to the prediction of the sample’s properties, needs careful consideration of factoring in differences derived from instruments, sampling, and analyst operations. The experimental design, from conception to data acquisition and beyond, changes significantly depending on the setting, with the research/academic environment having more control over the conditions (temperature, light, moisture, and motion) compared to industry. Also, sample conditions can dynamically change with the sample’s journey from industry to the laboratory, where most of the literature studies have been conducted. Some studies obtained samples under certain processing steps, while other studies scanned packaged samples [24], which will influence the model’s performance on newly presented samples. Naturally, the question arises as to whether the lack of integrating this variability into the experimental design can cause data (in)consistency problems and therefore very low model applicability.

For this purpose, it is evident that the calibration data should encompass as many sources of variation as possible for a well-designed study. Specifically, the incorporation of not only natural (e.g., seasonal, cultivar, environmental, food varieties or types, etc.) and processing (production parameters and storage) variability into a food sample but also other secondary types of real-world experimental variation (instrumental, sampling, and human factors) is necessary for the creation of universal and robust chemometric models and their transition to industrial online applications [25].

### 2.3. AI/ML/Chemometrics Pipelines

Over the last decade, both established and newly developed chemometrics techniques have been applied in food applications to tackle various emerging problems (Table 1). Moreover, chemometric approaches have been used for multi-analytical methods that are outside the field of non-destructive analysis, including untargeted HPLC [26], nuclear magnetic resonance [27], elemental metabolomics [28], mass spectrometry [29], and stable isotope analysis [30]. In this regard, there are specific chemometric techniques that have found wide applicability compared to others in the literature. In particular, principal component analysis (PCA), partial least-squares–discriminant analysis (PLS-DA), and partial least-squares regression (PLS-R) are by far the most widely used techniques in exploratory, classification, and regression analysis, respectively [31]. Nonetheless, these techniques are not always effective for solving non-linear analytical problems due to their assumption of linearity [32]. Additionally, the performance of the PLS-DA classifier is worsened by the increase in the number of model classes. This hinders its application to more complicated multi-class classification problems [33].

Regarding non-linear modelling methods, some studies have used SVM as a classifier, which aids in coping with multi-factorial problems using a mathematical concept called the kernel trick. However, this approach needs careful design consideration with the right number of support vectors and adequate hyperparameters. Also, SVMs are normally single outputs (meaning that they provide binary outputs for classification). Practically, the detailed setup of SVM algorithms is not mentioned very often in the published studies in which several commercial software packages were used directly for data analysis. Besides SVM, ANNs have found rapid development and wide applicability in various fields of science as a non-linear modelling method, thanks to their high predictability and practicability [80]. The ANN is an advanced calibration method, applicable in both classification and regression problems [81]. However, the ANN requirement for a large number of samples to achieve the appropriate modelling of the non-linear data and to reduce the risk of overfitting must be considered by anyone who uses this method, and this could, in theory, limit its application [82]. Another concern is the inherent complexity and lack of “explainability” (i.e., interpretation) of the ANN models, even if there are enough samples supplied to the algorithm. Addressing or mitigating these issues, certain types of ANNs, including FNNs, CNNs, and recurrent neural networks, when co-developed with computer and food scientists, have the potential to address complex food authentication analytical problems. On the other hand, the inevitable requirement of samples to cover the comprehensive variability will lead to the creation of huge databases with expensive proprietary software and hardware. Database management implies several challenges such as complexity, privacy, and storage problems [83]. This provides an additional challenge since important issues concerning computation time and complexity, exceeding memory demands, and low classification performance can be faced due to the increasing number of calibration samples [84]. Inherent to this issue, the use of batch learning methods requiring full recalibration after the addition of each new sample is inappropriate for the building of these universal chemometric models. Hence, there is a growing demand for accurate open-source chemometric tools to allow for the evolution or incremental learning of existing models and, at the same time, minimise computational and spatial costs [85]. In parallel, privacy and ethical concerns related to data sharing and user rights may hinder the development of large-scale general models when different proprietary databases are distributed across different institutions, private organisations, or countries. In this regard, the rise in federated learning (a decentralised ML approach to model building) as a novel privacy-preserving paradigm brings new opportunities to the sector [86].

To address the increasing complexity of real-world food applications, the building of more robust multi-class chemometric models is necessary [87]. For this purpose, the development of new chemometric methods that will enable the flexible modelling and representation of these multi-class datasets and their synergy with other pattern recognition techniques are two crucial requirements going forward with this challenge. Besides this, these methods should be highly sensitive to the detection of adulterants/contaminants at very low percentages and the simultaneous determination of multiple components in complex mixtures. Both are considered difficult real-world analytical problems.

### 2.4. Model Validation

Model validation is an important step in the construction of a chemometric model for assessing the accuracy, reliability, robustness, and predictive power of a model for unknown samples, optimising the model parameters, and preventing model overfitting or underfitting [88]. The phenomenon of overfitting is observed when a model has been trained with limited data and is dependent on these data and, therefore, has lost its generalisation ability [89]. Moreover, in underfitting, a model that has been created with too few components/variables or original examples (samples, in this case) cannot cover the variability in real-world data [90]. In addition, it is crucial to ensure that the same samples are not used to build the model and to evaluate the predictive power of the model [14]. Due to the importance of the validation step, two cross-validation schemes (“Leave-One-Out” (LOO) and k-fold cross-validation) are mainly used in the relevant literature [91,92]. LOO cross-validation produces overestimated results and potentially conceals overfitting when small validation datasets are used, resulting in classifying unrepresentative samples as part of the cross-validation set [93]. In addition, most published studies follow a single-laboratory validation strategy instead of an inter-laboratory validation method where the same experimental trial is performed and assessed in different laboratories to assess the transferability of the chemometric models.

Therefore, obtaining significant results requires not only meaningful analysis but also appropriate validation strategies, which are crucial in providing proof that the developed method has realistic and unbiased responses for future unknown samples [94]. Apart from cross-validation and external validation using independent testing samples/data, the application of inter-laboratory validation studies to the developed chemometric models to evaluate their prediction ability in other laboratories is essential. However, this is limited in food science studies due to the necessity of an adequate number of training samples incorporating the variability of multiple laboratories in different instrumental conditions [95]. This also has the potential to increase cross-lab collaboration, exploit federated chemometric models, and make data and model sharing a common practice, a practice that is currently lacking in the field [86].

### 2.5. Analyst Perspective

The human influence in decision-making is substantial. In theory, non-destructive rapid food analysis provides an unbiased, automatic, and ‘impersonalised’ result, removing the human element due to the high complexity of the data structure and the “black box” nature of the modelling methods. There are examples in industry where such offline instruments (FOSS Milkoscan) are in place. In these cases, the human is the user of the method.

On the contrary, in small-scale studies published in the literature, the analyst is a person with sufficient knowledge of data analysis and who is more involved in the process as both the user and the developer. Results from modelling could be reasonably adjusted or overfitted, and these adjustments may not always be in favour of the generality of the model. In some cases, one can argue that the same dataset can produce either mixed or controversial results depending on the analyst in charge [96]. Therefore, upholding integrity when dealing with spectral data is vital, especially in high-risk applications. Model validation is a critical strategy to mitigate the risk of producing overrated prediction performance. Debates may also arise concerning the size and proprietary nature of the experimental dataset. Therefore, open-source data and modelling methods can implicitly provide another form of validation, as human interference cannot dominate the distortion of the results.

## 3. How to Address the Challenges

### 3.1. Sensor/Hardware Development

Despite the achievements in analytical instrumentation over the last few years, the development of new types of cost-effective instrumental techniques combining spectroscopic techniques and other instrumental analytical techniques is a challenging task due to technological limitations (in optics, detector technology, etc.) [97]. In addition, the complexity of the data acquired increases exponentially [98]. The development of new non-destructive sampling procedures that will enable the frequent collection of representative and adequate samples, independently of their physical nature (solid, liquid, or gas), for highly accurate and reliable posterior analysis also presents a great challenge [99].

There is penetration of rapid, easy-to-use spectroscopy-based instruments in industry based on existing benchtop or portable sensor designs. The feed industry has already replaced chemical analysis for basic feed composition with near-infrared spectroscopy and chemometrics, and several analytical targets (total protein, fats, carbohydrates, etc.) have been tested [100]. Moreover, rapid methods developed using vibrational spectroscopy and chemometrics are already used in quality control/quality assurance (QC/QA) in the dairy industry, such as the Milkoscan^®^ and FoodScan^®^ from FOSS Instruments (Hilleroed, DK). Benchtop devices, such as the DA 7250 NIR analyser from PerkinElmer (Waltham, MA, USA), the TANGO FT-NIR from Bruker (Ettlingen, Germany), and handheld devices, e.g., MicroNIR OnSite-W from VIAVI Solutions (Milpitas, CA, USA), are designed specifically for analysis in the food and agriculture industries. To support these instruments and the analyses conducted, either the instrumental manufacturer or a third party provides calibration models. Besides this, third parties, such as AuNIR Ltd. (Towcester, UK) and Sagitto Ltd. (Cambridge, New Zealand), specialise in chemometric models that support the agri-food industry by developing, maintaining, and updating calibration models that extend the functionality of existing spectral instruments to provide more precise analysis. 

### 3.2. Dedicated Chemometrics Approaches for Further Analysis

As discussed, the current limitations of chemometrics methods are that present algorithms lack the capability to integrate varied datasets, which limits their effectiveness. A potential interim solution is the construction of an ensemble of models each trained on different datasets. This ensemble, functioning related to a decision support system, could collect insights from multiple models to deliver a comprehensive analysis.

Moreover, each spectroscopic instrument provides individual information for different food analyses, making assessing food products challenging, of which multi-sensor systems hold considerable promise [101]. For instance, NIR alone cannot holistically decipher the structural information of food products. Therefore, combining NIR with other techniques, such as MIR or SERS, can lead to an encompassing evaluation of food by integrating multiple information sources. Data fusion from complementary sensors has been employed for authentication and quality assessment, significantly boosting the performance of individual instruments [102,103]. Regarding HSI data, modelling and wavelength selection methods for spectral data are not entirely effective at solving the issue of 3D data. The spatial information of HSI also needs analysis, leading to the application of data fusion techniques to maximise the use of HSI data [103]. Data fusion achieves convincing results by separately evaluating the spectral and spatial information from HSI data. Kucha et al. [104] applied three data fusion methods to predict the intramuscular Fat content in pork samples, and data fusion resulted in a higher capacity for predicting this parameter.

Recent advances in neural network models have also marked a significant shift in interpreting complex chemical characteristics [105]. These models, particularly optimised for real-world applications in food analysis, reveal the potential field of artificial general intelligence (AGI) in food products. Despite their resource-intensive nature, these models promise a systematic approach to data analysis, pending the availability of sufficient computing power and annotated data to support their real-world application. Additionally, the suggestion to apply fuzzy-related and Bayesian-based models could potentially mitigate the opacity of NN and SVM, providing an interpretable analysis. A fuzzy system is a method for modelling and processing imprecise and vague information, which is characterised by membership functions that assign degrees of belonging to each variable [106]. Moreover, explainable and trustworthy AI are two recent and emerging topics that aim to address these challenges with the development of new algorithms. These proposed approaches would respond to several key issues and facilitate incremental and federated learning—concepts not exclusively pertinent to the food sector but universally recognised as challenges in diverse fields.

A critical observation in this paper is the necessity to design the chemometric pipeline prior to delving into modelling methods. Current studies often disproportionately emphasise sophisticated classification or regression techniques such as optimised decision tree models, neglecting the importance of a systematic chemometric process with simpler approaches. This process should span from preprocessing to the application of modelling methods. Regarding preprocessing, the integration of additional tools and methods is recommended. While PCA is a standard approach, it is not without its limitations. Moreover, reimagining the outputs of instruments like FTIR as time-series datasets opens possibilities for the application of decomposition techniques (without the Fourier transformation algorithm). These strategies might hold the key to refining the approach, underscoring the potential that lies in reevaluating and enhancing preprocessing stages before exploring new modelling schemes. To describe future rapid spectroscopic analysis, one could cite a message from the NIR 2023 conference: “There will be no causal interpretation without prediction validation and no prediction without attempted causal interpretation”.

The concept of multi-class and multi-task problems, involving more than two classes or objectives, further highlights the potential of using multiple models in parallel. By combining the outcomes of these models, an ensemble approach can be realised. This methodology aligns with the foundational principles of ensemble methods, underscoring their relevance and applicability in addressing complex challenges in chemometric analysis [107]. The multidisciplinary approach, encompassing algorithm development, preprocessing enhancement, and ensemble methods, presents a comprehensive strategy for advancing chemometric methodologies in the food industry and beyond [107].

### 3.3. Artificial Intelligence-Driven Methodology for Rapid Food Analysis

Several review papers have highlighted the importance of the sub-field of AI for rapid food analysis, including machine learning and deep learning, and focused on the implementation of ML/AI in a practical way [108]. The application of data-driven models, particularly different types of neural network models, in food manufacturing underscores a growing reliance on ML/AI to elucidate complex input–output relationships [109]. This data-centric approach could be beneficial for characterising systems where mechanistic knowledge is limited, thereby enhancing the predictive accuracy of quality and safety parameters for food products. However, the methodology of applying AI in future analytical studies lacks overviewing. On a theoretical level, the role of AI extends beyond modelling and simulation, offering benefits in the predictive maintenance of key factory equipment, the design of novel, cost-effective spectroscopy instruments, factory operational efficiency, and risk management [12]. Future studies under discussion should highlight the critical success factors for AI adoption in the food supply chain, including technology readiness, security, privacy, customer satisfaction, regulatory compliance, and the importance of information sharing among partners [110].

To meet real-world demand, various AI-driven approaches facilitate a collaborative environment where food scientists and computational experts can jointly tackle manufacturing challenges, directing the way for sophisticated multi-scale models that surpass traditional methods. This proactive stance is further exemplified in the development of leading indicators for food safety, where AI algorithms analyse behavioural data to anticipate potential safety issues, thus shifting the focus from reactive to preventive measures [111].

### 3.4. Model Validation

In the realm of chemometrics, the importance of robust data validation methods like k-fold cross-validation cannot be overstated. However, the effectiveness of these methods hinges on the availability of sufficient and high-quality data, as well as its ground truthing to facilitate training and evaluation of the chemometric models. A key aspect of ensuring data quality involves the detection and removal of outliers. Additionally, the creation of “synthetic” datasets has emerged as a viable solution to enhance the performance of classifiers, especially when dealing with limited or skewed data [112]. Besides removing observed outliers, data augmentation is a disruptive method used to enhance model robustness for developing and validating rapid food analysis methods, involving artificially enlarging the training dataset by creating modified versions of the existing datasets [113]. Data augmentation fundamentally enhances model validation: a more diversified dataset is used for training; the occurrence of overfitting is minimised; the training process becomes more efficient by gathering more information at each epoch. As a result, there is the enhancement of the model’s generalisability with exposure to increasing numbers of datasets [114]. In particular, data augmentation is carried out by integrating noise, such as Gaussian noise, or in combination with ensemble methods [112]. Additionally, there are more innovative techniques employing deep learning for data augmentation, including generative adversarial networks (GANs) and semi-supervised generative adversarial networks to create new data points [114]. In practice, data augmentation presents itself as an attractive low-cost strategy for robust temperature compensation in NIR calibration and prediction, challenging traditional methods that integrate non-relevant spectral variations by measuring samples under different conditions [115]. Two types of approaches have been proposed: augmenting the calibration matrix with simulated noise and a correction method to eliminate non-relevant variations from new spectra. Continuous development of data augmentation is a related framework for chemometric analysis to enhance the classification results; synthetic spectroscopic data from vegetable oil samples were used to assist the model’s prediction performance without the need for extensive ‘real’ sample datasets [112]. This study highlighted its potential in addressing challenges related to sample variability and instrument differences in spectroscopic analysis. Regarding the deep learning approach, a GAN was used for predicting the oil content of a single maize kernel by generating artificial spectra after many iterations, resulting in very promising results [116]. 

Overall, the integration of such methodologies into a standardised workflow is crucial for maintaining data integrity in chemometric analysis. This standardisation arguably minimises manual interference and prevents the generation of overly optimistic results stemming from unfit analytical strategies. From the perspective of utilising spectroscopy instruments in the industrial production line, the addition of historical data to the regular validation process could provide a forecast on operations and the decision-making process in a timely manner. The integration of blockchain technology for data sharing in validating data analysis steps represents innovative approaches to ensuring data transparency and reliability [9]. Moreover, the establishment of open repositories for spectroscopic data can democratise access to valuable datasets, further boosting research outputs. 

## 4. Conclusions

Nowadays, there is an increasing need for rapid, easy, non-destructive, and low-cost analytical solutions with minimal sample preparation and accurate results in the food field to control the quality and safety of raw materials and intermediate and final products and identify any issues such as food adulteration (Figure 3). Vibrational spectroscopic methods are untargeted analytical techniques capable of meeting these demands, at the cost of producing complex multidimensional data, containing plenty of useful information for analysis but whose interpretation is commonly a difficult, laborious, and time-consuming task. Chemometrics offers a wide spectrum of data analysis methods that can enable the deep and extensive analysis of the acquired spectra, as well as its automation, by allowing their elucidation, analysis, and further modelling thanks to its interdisciplinary nature. However, there are some practical issues related to the use of vibrational spectroscopy associated with chemometrics that limit its online industrial applications. Generally, chemometric methods demand a spectral library of high quality and representative spectra to build accurate, sensitive, robust, generalised chemometric models. Concerning sample representativity, vibrational spectroscopy requires sample homogeneity over the sampling area to produce a representative spectrum of an examined sample. This fact limits its application to heterogeneous and complex food samples where better sampling techniques are needed. Moreover, conducting the analysis in a laboratory is different than in a real-world industrial environment. A laboratory has well-controlled conditions for sample handling, preparation, and subsequent analysis, with little variation affecting data acquisition, whereas the same analysis in the field will create more noise, i.e., more variation that the models have not been exposed to during development [117]. This means that spectroscopic acquisition can be affected due to variations in experimental or sample conditions. For this reason, it would be preferable if the developed model incorporates all the possible sources of sample variation, such as natural and experimental variation. Nonetheless, the inclusion of all relevant sources of sample variation seems impractical due to limited resources since it requires time-consuming and costly processes as well as exceeding computer processing and memory demands or fulfilling the required privacy and legal constraints. Moreover, apart from the practical limitations, the complexity of the developed chemometric model may be increased rapidly, and low classification performance may be faced due to the increasing number of training samples and model classes.

Except for the issues related to the production of realistic solutions for food analysis, current classification and regression techniques meet challenges when they are used for the detection of adulterants at low concentration levels and the characterisation of complex food blends, where many compounds must be identified and quantified [118]. This can be explained by the difficulty associated with these analytical problems due to the subtle spectral differences between authentic and adulterated mixtures, as well as the overlapping peaks of compounds in mixtures. Thus, even though chemometrics offers automatic results, current methods are limited for low-level adulterations and multi-component analysis and current models are not general enough and often overfitted when and if extended to real-world applications. 

On the other hand, there is no significant “year-over-year” or even “decade-over-decade” scientific progress in the efficiency of these algorithmic methods to tackle even more complex food analysis problems. If one considers the example of the speech recognition and natural language processing research area, the rise in powerful novel ML algorithms such as deep learning systems and follow-up concepts such as reinforcement learning, federated learning, trustworthy AI, etc., have “transformed the scene” leading to digital assistants on our phones (Siri and Google Assistant), AI dominance in the game Go (DeepMind) [119], and, of course, the recent introduction of large language models trained on generative pre-trained transformers (OpenAI and ChatGPT). It remains to be seen, however, as to how these hyper-advanced methods will be translated to the field of chemometric-based rapid food analysis. At the top, there is a limited amount of research, produced in academic studies, that finds its way into industry and the official methods of regulatory bodies, which highlights a gap in technology transfer as well as the applicability of the results.

## Figures and Tables

**Figure 1 foods-13-00846-f001:**
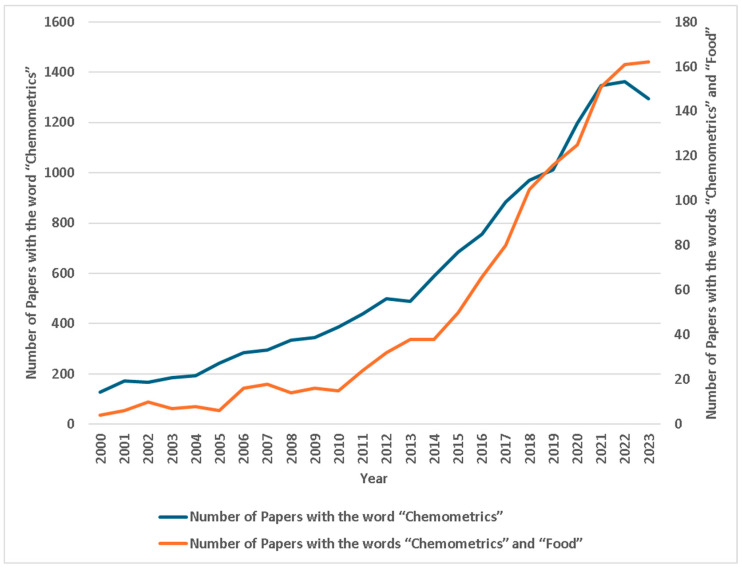
Rise of research papers per year with the word “chemometrics” (main axis) and a combination of “chemometrics” and “food” (secondary axis).

**Figure 2 foods-13-00846-f002:**
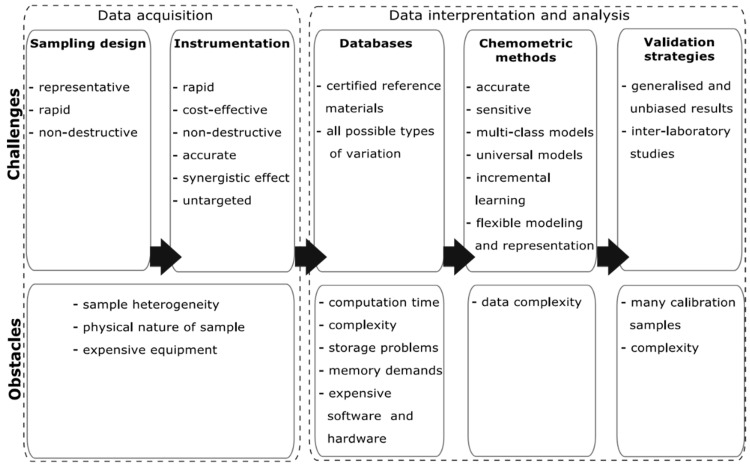
Modern research challenges in rapid food analysis.

**Figure 3 foods-13-00846-f003:**
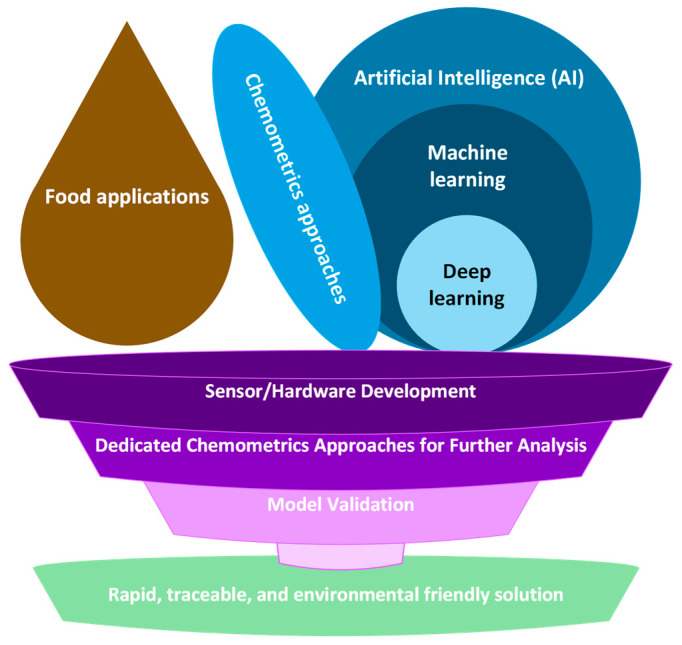
Overview of AI-driven non-destructive spectroscopic tools for rapid food analysis.

**Table 1 foods-13-00846-t001:** Development of chemometrics in food applications using non-destructive spectroscopic tools.

Chemometrics	Technique	Applications	Performance (as Reported)	Refs
LDA	NIR	Alcohol degree of Chinese liquor	R^2^ = 0.96	[34]
LDA	Raman	Adulterants (fructose corn syrup and maltose syrup) in honey	Accuracy = 91%	[35]
QDA	HSI	Detection of anthracnose in mango fruits	KAPPA = 90%	[36]
KNN	NIR	Geographic origin discrimination of millet	Discrimination rate = 0.99	[37]
KNN	Raman	Detection of adulteration of extra virgin olive oil	Classification rate = 0.79	[38]
KNN	MIR	Monitoring of soluble pectin content in orange juice	Accuracy = 85%	[39]
SIMCA	NIR	Identification of the types of fat added to feed	Sensitivities = 100%	[40]
SIMCA	Raman	Detection of milk powder adulteration	Accuracy = 97%	[41]
PCR	Raman	Honey adulteration	Accuracy = 96.54%	[42]
MLR	NIR	Soluble solids content of tea soft drinks	R^2^ = 0.98	[43]
MLR	MIR	Mycotoxin deoxynivalenol (DON) in wheat	R^2^ = 0.99	[44]
PLS	NIR	Aflatoxigenic fungal contamination in rice	R^2^ = 0.67	[45]
PLS	SERS	Identification of food processing bacteria	Accuracy = 99%	[46]
PLS	MIR	Quality characteristics in pomegranate kernel oil	R^2^ = 0.91	[47]
PLS	THz	Detection of moisture content for *Ginkgo biloba* fruit	R^2^ = 0.78	[48]
PLS	HSI	Identification of sun-dried and sulphur-fumigated herbals	Sensitivity = 96%	[49]
SVM	NIR	Adulteration of food products (extra-virgin-olive oil, honey, milk, and yogurt)	Accuracy = 0.90–1.00	[50]
SVM	Raman	Adulteration of extra virgin olive oil	R^2^ = 0.99	[51]
SVM	MIR	Discrimination of wild Paris Polyphylla Smith var. yunnanensis	Accuracy = 87%	[52]
SVM	THz	Use in dried tangerine peels	Accuracy = 94%	[53]
SVM	HSI	Authentication of Theobroma cacao bean hybrids	Prediction error = 3.8–23.1%	[54]
DT	NIR	Poultry quality classification	Precision = 0.74	[55]
DT	Raman	Identification of foodborne pathogenic bacteria	Correct recognition ratio = 0.98	[56]
DT	MIR	Rapid screening of aflatoxin-contaminated peanut oil	Sensitivity = 100%	[57]
DT	THz	Moisture content in fruits and vegetables	Accuracy > 94%	[58]
DT	HSI	Evaluation of subjective tea quality	R^2^ = 93%	[59]
RF	NIR	Determination of the food dye indigotine in cream	R^2^ = 0.94; RPD = 4.09	[60]
RF	Raman	Classification of milk products from cows, buffalos, and goats	Accuracy > 94%	[61]
RF	MIR	Identification of the geographical origin of black tea	Accuracy = 100%	[62]
RF	THz	Identification of rice powder mixtures	Accuracy = 98%	[63]
RF	HSI	Quantification of *Clostridium sporogenes* spores in food products	Accuracy = 80%	[64]
FNN	THz	Prediction of gelatin of various animal origins	Accuracy = 100%	[65]
FNN	HSI	Measurement of firmness and soluble solids content for apples	R^2^ = 0.76, 0.79	[66]
ANN	NIR	Measurement of carbohydrates and moisture in rice	R^2^ = 0.98, 0.97	[67]
ANN	Raman	Rapid analysis of sugars in honey	R^2^ > 0.96	[68]
ANN	THz	Prediction of the freshness of pork	RMSEP = 9.9%	[69]
ANN	HSI	Prediction of moisture content in Lonicerae Japonicae Flos	RPD = 4.42	[70]
CNN	NIR	Determination of the soluble solid content of crown pear	R^2^ = 0.96	[71]
CNN	SERS	Quantification of thiram and pymetrozine in tea	R^2^ = 0.99, 0.98	[72]
CNN	MIR	Identification of sugar adulteration in honey	Accuracy = 100%	[73]
CNN	THz	Classification of wheat grain varieties	Accuracy = 98%	[74]
CNN	HSI	Quantitative adulteration in Atlantic salmon	R^2^ = 0.99	[75]
ELM	NIR	Detection of fennel origin	Accuracy = 100%	[76]
ELM	Raman	Identification of infant rice cereal	Accuracy = 99%	[77]
ELM	THz	Identification of adulterated rice seeds	Accuracy = 100%	[78]
ELM	HSI	Prediction of the cadmium content in rape leaf	R^2^ = 0.98	[79]

Note: LDA: linear discriminant analysis; QDA: quadratic discriminant analysis; KNN: K-nearest neighbour; SIMCA: soft independent modelling of class analogy; PCR; principal component regression; MLR: multiple linear regression; PLS: partial least squares; SVM: support vector machine; DT: decision Tree; RF: random forest; ELM: extreme learning machine; FNN: feedforward neural network; ANN: artificial neural networks; CNN: convolutional neural network; NIR: near-infrared; MIR: mid-infrared; SERS: surface-enhanced Raman spectroscopy; THz: terahertz; HSI: hyperspectral imaging; R^2^: coefficient of determination; RPD: ratio of performance to deviation; RMSEP: root mean square error of prediction.

## Data Availability

The original contributions presented in the study are included in the article, further inquiries can be directed to the corresponding author.

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
