# Peer review of "Challenges in the Use of AI-Driven Non-Destructive Spectroscopic Tools for Rapid Food Analysis"

_foods, 2024, doi:10.3390/foods13060846_

Round 1

Reviewer 1 Report

Comments and Suggestions for Authors

In this manuscript the authors have evaluated the methods used in chemometric-based studies. They pointed out the analytical techniques and current challenges. They also give practical suggestions to enhance the technical approach in future studies and offer innovative solutions. While the manuscript is well-written and covers a critical concept, I would like to offer the following comments for improvement.

·       Lines 27-28: Consider providing a more comprehensive) description for targeted analysis. While the following sentences enhances the description, the first sentence could be further refined.

·       Lines 112-114: The intended meaning of this sentence is unclear. Presumably, you are suggesting that, due to the sample size in chromatographic techniques, the result may only represent a specific portion of the sample. However, such as with FTIR crystal that are so small, homogeneity is crucial, but they are still applicable for routine analysis

·       Line 261: a reference note is forgotten.

·       Lines 288-297: This paragraph requires improvement. It would be beneficial to include various industry examples beyond just the feed and dairy industry.

Comments on the Quality of English Language

It is a well written and easy to understand manuscript. 

Reviewer 2 Report

Comments and Suggestions for Authors

The manuscript offers valuable insights into the challenges and opportunities in chemometrics for food analysis. With some minor revisions, including improving references, enhancing the conclusion section, and further discussing data sharing and reproducibility, the manuscript has the potential to make a significant contribution to the field.

Specific comments:

References: While the manuscript references relevant literature, some citations appear to be incomplete (e.g., "[ref]"). Ensuring all references are properly cited and listed would improve the academic rigor of the manuscript.

In-text Citations: The manuscript would benefit from more consistent and standardized in-text citations. Some statements lack citation to support the claims made, which could undermine the credibility of the arguments presented. Ensuring that all statements are properly supported by references would strengthen the academic rigor of the manuscript (E.g. lines 309-310: "Recent advances in neural network models have marked a significant shift in interpreting complex chemical characteristics"; lines 362-365: "The complexity of modern food analysis requires an unprecedented level of sophistication"; lines 402-403: "Current classification and regression techniques show low performance when used for the detection of adulterants at low concentration levels";  lines 388-390: "The laboratory has well-controlled conditions in sample handling, preparation, and subsequent analysis with little variation affecting the data acquisition"; lines 356-357: "The integration of blockchain technology for data sharing represents an innovative approach to ensuring data transparency and reliability").

Incorporation of Visual Aids: Consider incorporating figures, diagrams, and tables to visually represent key data and concepts discussed in the manuscript. Visual aids can improve the presentation of information and enhance reader understanding.

Conclusion: The conclusion section could be strengthened by summarizing key findings and providing clear recommendations for future research directions.

Data Sharing and Reproducibility: The manuscript briefly touches upon the importance of data sharing and reproducibility but could delve deeper into these topics, discussing specific initiatives or recommendations for improving transparency and reproducibility in chemometric studies.

Reviewer 3 Report

Comments and Suggestions for Authors

AI-assisted spectroscopic techniques for non-destructive testing of food quality have been widely used, which is one of the key technologies in the food field. This manuscript provides an overview of the relevant applications of AI-assisted spectroscopic detection techniques in the field of food from the perspectives of chemometric methods, experimental design, and model construction. The manuscript still needs to be supplemented to a great extent to improve its content.

1. The advantages and necessity of AI for the food field should be added in more detail.

2. There are a large number of applications of chemometrics in the field of food, so some typical examples should be summarised to improve the content of the manuscript from a methodological point of view.

3. Relevant data for the year 2023 should be added in Figure 1.

4. As a review article, the number of graphs and tables in the text is very small, which is a part that must be further improved.

5. Currently, there are many similar review articles in the field of food, and the authors should compare them vertically to highlight the necessity of this manuscript.

Comments on the Quality of English Language

Moderate editing of English language required.

Round 2

Reviewer 3 Report

Comments and Suggestions for Authors

This manuscript can be accepted for publication in the current version.